# Quantitative Metabolomics to Explore the Role of Plasma Polyamines in Colorectal Cancer

**DOI:** 10.3390/ijms24010101

**Published:** 2022-12-21

**Authors:** Donatella Coradduzza, Caterina Arru, Nicola Culeddu, Antonella Congiargiu, Emanuela Gigliola Azara, Antonio Mario Scanu, Angelo Zinellu, Maria Rosaria Muroni, Vincenzo Rallo, Serenella Medici, Ciriaco Carru, Andrea Angius, Maria Rosaria De Miglio

**Affiliations:** 1Department of Biomedical Sciences, University of Sassari, 07100 Sassari, Italy; 2Institute of Biomolecular Chemistry, National Research Council, 07040 Sassari, Italy; 3Department of Medicine, Surgery and Pharmacy, University of Sassari, 07100 Sassari, Italy; 4Institute for Genetic and Biomedical Research (IRGB), Department of Biomedicine, National Research Council, 09042 Monserrato, Italy; 5Department of Chemical, Physical, Mathematical and Natural Sciences, University of Sassari, 07100 Sassari, Italy; 6Control Quality Unit, Azienda-Ospedaliera Universitaria (AOU), 07100 Sassari, Italy

**Keywords:** colorectal cancer, biomarker, polyamine, agmatine, inflammatory indexes

## Abstract

Colorectal cancer (CRC) is one of the major public health and socio-economic problems, which management demands the development of non-invasive screening tests. Assessment of circulating polyamines could be a valuable tool, although analytical problems still preclude its clinical practice. We exploited ultra-high-resolution liquid chromatography and mass spectrometry, as a highly sensitive and innovative method, to profile eleven polyamines, including spermine and spermidine with their acetylated forms. These data together with an evaluation of the inflammatory indexes might represent suitable biomarkers for the identification of CRC patients. The statistical models revealed good discrimination in distinguishing CRC patients from healthy subjects. The plasma assessment of ornithine and acetylspermine, as well as lymphocyte/platelet ratio, revealed helpful information on the progression of CRC. The combined profiles of circulating polyamines and inflammatory indexes, together with the application of an innovative technology, could represent a valuable tool for discriminating patients from different clinical groups.

## 1. Introduction

Colorectal cancer (CRC) is a major public health and socio-economic problem. It is the third most commonly diagnosed cancer and the second leading cause of oncological death worldwide [1]. About twenty percent of CRC patients already have metastases at diagnosis, and metastatic CRC (mCRC) is mostly an incurable disease [2]. The five-year survival rate is 90% at stage I, with a severe decrease to about 10% at stage IV [3]. Despite advances in the knowledge of CRC biology and therapeutic improvements, it is still one of the hard-to-treat cancers due to the high frequency of metastases and drug resistance. The identification of K-RAS and N-RAS gene mutations is a widely-accepted molecular test in mCRC clinical treatment decisions [4]. The BRAF V600E mutation has been considered a good prognostic biomarker to identify patients who may develop an aggressive clinical outcome [4]. In mCRC patients, a connection between MSI-high tumor and a remarkable response to immune-checkpoint blockade with anti-PD1 therapy has been established [5]. In developed countries, the decrease in cancer mortality is attributable to population-based screenings and therapeutic improvements [6]. Nevertheless, the increasing incidence of CRC in young people [7] indicates that there is a lack of highly specific diagnostic tests (i.e., CEA, CA19-9, and CA15-3) to enable primary prevention of this disease, because merely considering the fecal occult blood test is not sufficient. The diagnosis and management of CRC remain an overwhelming challenge. The gold standard for CRC diagnosis is still colonoscopy, an invasive and troublesome test that discourages patient involvement. It is a very expensive exam for the healthcare system and requires highly specialized medical personnel [8].

It is important to expand the use of non-invasive methods and employ different biological materials, such as blood, urines, and faeces, to increase patient compliance, and also to develop robust biomarkers for population monitoring [9]. 

Systemic host inflammatory response is a hallmark of cancer, which is implicated in the development, progression, and prognosis of many cancers, including CRC [10]. Hematological inflammatory indexes and an increase in some of them [neutrophil–platelet to lymphocyte ratio (SII), neutrophil to lymphocyte ratio (NLR), platelet to lymphocyte ratio (PLR), and monocyte to lymphocyte ratio (MLR)] has been clearly linked to higher recurrence risk and worse outcomes in CRC [11].

Inflammatory indexes and metabolites may allow the identification of fingerprints of neoplastic tissue biochemistry that correlate with cellular phenotypes [12]. Several authors have highlighted the involvement of polyamines in tumor development. Polyamines are polycationic alkylamines that are ubiquitously distributed in mammalian cells, particularly they are highly produced in rapidly growing cells. They are synthesized from ornithine by an initial decarboxylation step to putrescine and catalyzed by ornithine decarboxylase (ODC); this is followed by an aminopropyl transfer reaction via spermidine synthase (SRM) and spermine synthase (SMS), which generates spermidine and spermine, respectively. Under physiological conditions, due to the plasticity of their charge distribution, polyamines bind several negatively charged molecules, including nucleic acids, ATP, proteins, and acid phospholipids, and perform critical biological functions, such as DNA stabilization, nucleic acid and protein synthesis, gene regulation, cell growth, proliferation, differentiation, migration, development, and immunological responses [13]. They are also involved in cell adhesion and extracellular matrix repair, and in specific signaling processes [14]. Cellular polyamine levels are ruled by a dynamic balance between polyamine biosynthesis and catabolism, which could change during carcinogenesis in a way that favors cancer development [14]. Many studies underline the role of polyamines as promising biomarkers in cancers by analyzing their metabolic profiles on different biological fluids [15,16,17]. Changes in the profile of polyamines in urine can discriminate CRC patients from healthy ones or those with benign lesions [18,19]. Agmatine, a polyamine obtained via arginine decarboxylase (ADC) that is present in mammalian systems [20], hinders the polyamine pathway by blocking ODC activity, and inhibits polyamine uptake [21]. This process induce tumor cell proliferation repression [22]. Above all, the antiproliferative effect of agmatine has aroused interest as a new therapeutic choice in the treatment of tumors. 

The aim of this study is to test whether the levels of circulating polyamines in plasma and the inflammatory indexes can discriminate CRC patients from healthy ones, and to relate the clinical-pathological features of CRC to biochemical signatures.

## 2. Results

Appendix A shows the clinical-pathological features of the 50 CRC patients (33 men; 17 women) included in this study. The median age was 67.74 years (age range: 46–84 years), with 31 (62.0%) patients being older than 65 years old. Thirty-five (70.0%) tumors affected the left colon. Tumor staging was pT1 in 5 (10.0%) cases, pT2 in 11 (22.0%) cases, pT3 in 28 cases (56.0%), and pT4 in 4 cases (8.0%). Lymph node status was divided into 26 pN0 (52.0%), 15 pN1 (30.0%), and 7 pN2 (14.0%). Moreover, 2.0% of tumors were at stage 0, 20.0% at stage I, 26.0% at stage II, 34.0 at stage III, and 14.0% at stage IV; 8.0% of CRCs were G1, 70% were G2, and 20.0% were G3. Seven patients (14.0%) showed metastasis at diagnosis.

### 2.1. Biochemical Parameters

Table 1 shows all analyzed biochemical parameters. We found a statistically significant difference between the CRC patients and the healthy subjects for the majority of hemochrome-cytometric examination values, except for WBC and PLT, and for the inflammation-related parameters, except for MPR. The CRC patients show a decrease in HGB, RDW, MPV, and lymphocytes values, and an increase in MCV, neutrophil, and monocyte values compared to the healthy subjects. HGB/RDW shows a significant decreasing trend in the CRC patients. The inflammatory indexes are particularly high in the patients compared to the healthy subjects (Table 1 and Appendix A). Table 2 shows the correlation between the inflammatory indexes and clinical-pathological CRC features. Statistically significant differences are observed between the inflammatory indexes, MPR, PLR, and HGB/RDW ratio, and tumor localization, and between NLPR and histologic grade. MPR and HGB/RDW ratio increase prevalently in left tumors and PLR in right tumors, while NLPR increases in low-grade tumors (Table 2 and Appendix A).

### 2.2. Plasma Levels of Polyamines, Related Amino Acids, and Metabolites

Table 3 shows the levels of plasma polyamines and related amino acids (arginine and lysine). All parameters show significant differences between the two groups. The CRC patients present a significant increase in arginine, cadaverine, lysine, ornithine, putrescine, acetylputrescine, spermidine, acetylspermidine, spermine and acetylspermine levels compared to healthy ones. An opposite trend is observed for agmatine level (Table 3, Appendix A). Table 4 shows the correlation between polyamine levels and clinical-pathological features in the CRC patients. Statistically significant differences are observed between ornithine and histologic grade, between acetylputrescine and sex, and between acetylspermine and tumor stage. Specifically, ornithine predominantly increases in low-grade tumors, acetylputrescine increases in female patients, and acetylspermine increases in low-stage tumors (Table 4, Appendix A).

### 2.3. Multivariate Analysis

To better understand the role of polyamines in CRC, we performed a multivariate analysis on the patients’ plasma data integrated with clinical data. In Figure 1, the multivariate analytical method, using the PLS-DA (partial least squares discriminant analysis), shows excellent discrimination between the CRC patients and the healthy subjects. The two groups form two distinct clusters. The supervised analysis was performed by applying OPLS-DA (orthogonal partial least squares discriminant analysis), which implies a rotation of the corresponding PLS-DA models and simplifies the information into a predictive component while maintaining the same predictive capacity. In Figure 2, the VIP obtained from the OPLS-DA model highlights the contribution of the analytes in discriminating the two groups. Among the polyamines, spermidine, acetylputrescine, acetylspermidine, cadaverine, acetylspermine, agmatine, and ornithine, together with RDW and PLT, are the variables that clearly discriminate these two groups. To avoid model overfitting, the OPLS-DA models were validated with a 300-fold figure permutation (Figure 3). The resulting regression lines show an R2 intercept at 0.0519 and a Q2 intercept at −0.291, indicating a valid model [23].

## 3. Discussion

The present study aims to identify a peculiar biological signature in the metabolism of the amino acids, arginine and lysine, as a form of fingerprinting for CRC. The quantitative metabolomic analysis was performed using a highly sensitive and innovative method, by coupling UHPLC and HRMS, which has the advantage of extracting target ions from the total ion chromatogram. This protocol is capable of detecting and quantifying even underrepresented polyamines in complex matrices [23]. Furthermore, the simultaneous analysis of eleven plasma targets, including their acetylated forms, has made it possible to identify some polyamines that differ between the cancer patients and the healthy subjects.

Our data show that the concentrations of the acetylated polyamines, non-acetylated polyamines, arginine, cadaverine, and lysine are statistically different among the comparison groups and are sharply increased in the CRC patients. The multivariate analysis shows a net clustering into two cohorts, primarily influenced by spermidine, acetylputrescine, acetylspermidine, cadaverine, acetylspermine, ornithine, and agmatine.

Our results show a statistically significant difference for acetylputrescine concentration between females and males, as well as an increase in ornithine predominantly in low-grade tumors and an increase in acetylspermine in low-stage tumors. Considering that polyamines are particularly highly produced in rapidly growing cells, our results suggest that ornithine and acetylspermine are specifically related to tumor progression. The increase of polyamine in polyps and preneoplastic colorectal tissues [24] supports our finding.

In CRC patients, Linsalata et al. revealed a decreased activity of polyamine oxidase, the enzyme responsible for acetylspermine and acetylspermidine catabolism [25]. Difluoro-methyl-ornithine, an irreversible inhibitor of ornithine decarboxylase, has suppressive activities on colorectal adenomatous polyps. This highlights a stimulation of intracellular polyamine growth in the early stage of CRC tumorigenesis, which could be useful as a target for chemoprevention in high-risk populations [26]. Our data suggest that ornithine and acetylspermine plasma assessment could be used as biomarkers to distinguish early and advanced CRC.

Spermine and acetylspermine are able to differentiate cancer according to different stages and histologic grade in different tumor types [18]. The profile of our plasma analytes not only provides information on general metabolic changes, but the single analytes also give a differential global profile.

Polyamine levels are elevated in patients with different cancer types and, considering their cellular function, they might be linked to cancer development. Furthermore, high polyamine levels are associated with the progression of neuroblastoma; prostate, lung, breast, gastric, and hepatocellular carcinoma; and lymphoma [15,16,27,28,29,30]. Polyamines are crucial for determining tumor immune privilege that stimulates the growth of cancer cells via the excretion of spermine and its metabolites [31]. In addition, polyamines accelerate chemoresistance acquisition to 5-fluorouracil and paclitaxel in CRC and breast cancers [32,33].

In recent years, the role of gut microbiome and its dysbiosis is becoming clearer in cancer development [34]. The gut microbiota generates various metabolites following anaerobic fermentation of undigested dietary parts. Polyamines are the main gut microbiota metabolites with pro-carcinogenic effects [35]. The interchange between the microbiota and tissues generates a tumor microenvironment in which polyamine metabolism, content, and function might be markedly distorted on the basis of microbiota composition, dietary polyamine availability, and tissue response to its surrounding microenvironment, influencing potential cancer development [36]. A previous metabolomics study displayed a positive feedback loop between the microbiota and the host, in which polyamines generated by CRC cells stimulate the growth of bacterial biofilms, and this produces polyamines to cause cancer cell growth [37]. Our results identify a strong decrease in agmatine levels in the CRC patients compared to the healthy ones. The observed agmatine level trend in the plasma of CRC patients is present in different tumors [38]. Agmatine is a polyamine formed by arginine decarboxylation. Arginine regulates different biological processes, such as cell growth, and becomes a limiting factor in the conditions of a rapid cell turnover, as in cancer tissues, but it is also required for the immune destruction of malignant cells [39]. Arginine deprivation therapy is being investigated as an adjuvant therapy for cancer. Agmatine is irregularly distributed in organs and tissues and has a dual behavior. It restricts polyamine synthesis by reducing the activity of ODC and inhibits the uptake of polyamines. These effects lead to the suppression of tumor cell proliferation in vitro [40], as well as in in vivo models, by affecting the polyamine pathway [41].

Routine CBC analysis identifies diagnostic and prognostic markers for cancer patients. In our study, the inflammatory indexes analyzed, such as AISI, SII, SIRI, MRL, NLR, NLPR, DNLR, and PLR, differ between the two patient classes in a statistically significant manner and appear significantly increased in the CRC patients. These values, even singularly, can be considered as circulating diagnostic markers. The correlation between the inflammatory indexes and clinical-pathological features in CRC shows that NLPR increases in low-grade tumors. Passardi et al. demonstrated that NLR is a prognostic independent factor of disease-free survival (DFS) and overall survival (OS). Patients with low NLR who were treated with chemotherapy (CT) plus bevacizumab had a higher DFS than those treated with CT alone [42].

Our patients with CRC show a strong reduction in HGB, RDW, and HGB/RDW ratio. HGB/RDW is a predictive factor already evaluated in other tumor types. Low HGB levels may reflect malnutrition and host immune status and, as with RDW, may indicate poor treatment tolerance. Anemia before treatment is an unfavorable predictor in patients with nasopharyngeal carcinoma, head and neck carcinoma, cervical carcinoma, esophageal squamous cell carcinoma, and gastrointestinal carcinoma [43,44,45,46,47,48]. RDW is closely associated with unfavorable outcomes in various diseases, such as cardiovascular, pulmonary, and hepatic diseases [49,50,51], and with the clinical features and prognosis of oncological diseases [52,53]. It is known that both HGB and RDW are affected by various pathological conditions, whereas the HGB/RDW ratio is an oncological predictive factor. HGB/RDW is an inexpensive and feasible parameter for CRC, and we suggest including it to facilitate the management of these patients.

We acknowledge that our study has some limitations related to its retrospective nature and sample size. The results need validation in a larger cohort, but the method is promising as a tool to support diagnostic stratification. Extending the study to polyamines in plasma samples from patients at various pre-tumor, tumor, and/or post-disease progression or recurrence stages could identify a prognostic biomarker, supporting patient stratification, clinical course, and identification of very high-risk patients. Future studies need to explore the clinical significance of blood parameters, such as HGB/RDW ratio, and validate their use. The hope is to discover the key to new biomarkers. The results suggest that a single biomarker does not reflect the complete picture of this disease, but a broader set of biomarkers can build a predictive model in accordance with personalized medicine practices.

## 4. Materials and Methods

### 4.1. Chemicals and Methods

The chemicals for the analysis were purchased from Sigma-Aldrich (St. Louis, MO, USA), and water for ultra-high-performance liquid chromatography–high-resolution mass spectrometry (UHPLC/HRMS) was purchased from Thermo Fisher Scientific (Fair Lawn, NJ, USA) [16]. The measurement of plasma levels of polyamines was performed using an UPLC Ultimate 3000 (Thermo Fisher Scientific) system equipped with a HESI-II electrospray source to a Q-Exactive-Orbitrap™-based mass spectrometer (Thermo Fisher Scientific). Chromatographic separation was carried out on a C18 column of the Gemini C18 (Phenomenex, Torrance, CA, USA), 100 mm × 2 mm, particle size 3 µm, and the column was held at 37 °C. Chromatographic separation was achieved with gradient elution using a mobile phase composed of 0.05% heptafluorobutyric acid (HFBA) in water and 0.05% HFBA in methanol, as described in Ran Liu et al. and modified by Coradduzza et al. [54,55].

### 4.2. Plasma Colorectal Carcinoma Sample Collection

This study follows the Declaration of Helsinki’s guidelines and was approved by the Azienda Sanitaria Locale Sassari Bioethics Committee (n. 2032/CE, 13/05/2014). All patients gave written informed consent. This study examined 50 consecutively anonymized CRC patients treated by surgical resection at the Unit of Surgery of the University of Sassari, between May 2019 and September 2021, and 52 healthy volunteers from the transfusion center of the Health Protection Agency of Sassari. All tumors were reviewed by two experienced pathologists and categorized according to the current WHO classification [56]. Patients with infectious diseases or receiving medical treatments or neoadjuvant therapies that could alter molecular analyses were excluded. Coexistence of clinical or histopathological findings with regard to inflammatory bowel disease (IBD; i.e., ulcerative colitis or Crohn’s disease) was an exclusion criterion from the study; in particular, no case with CRC had IBD and no patient in the healthy group was affected by chronic bowel disease.

Whole blood samples were collected before surgery from each eligible CRC patient and from the healthy subjects. Whole blood was centrifuged at 3000 rpm for 15 min and all plasma samples were stored at −80 °C. Analysis was performed as described in Coradduzza et al. [15,55]. Complete blood count (CBC) analysis, in CRC patients and healthy patients included in this study, was collected prior to surgery.

### 4.3. Statistical Analysis

The D’Agostino-Pearson’s test was performed to evaluate variable distribution [57]. Data are expressed as mean values (mean ± standard deviation (SD) or median values (median and interquartile range (IQR)). Between-group differences of continuous variables were compared using the Student–Newman–Keuls test or the Kruskal–Wallis test, as appropriate, assuming a *p*-value < 0.05 as statistically significant. A supervised analysis was carried out by applying the orthogonal partial discriminant analysis of the minimum square (OPLS-DA), representing a rotation of the corresponding PLS-DA models and simplifying the information into one predictive component only while maintaining the same predictive capacity [58]. To avoid model overfitting, the corresponding PLS-DA models were validated by 300-fold permutation tests [59]. The prediction strength of the model was evaluated using “Leave out” analysis [60]. Variable Importance Parameter (VIP) values were used to assess the overall contribution of each X variable to the model, summed over all components and weighted according to the Y variation accounted for by each component. The number of terms in the sum depends on the number of PLS-DA components found to be significant in distinguishing the classes. The Y-axis indicates the VIP scores corresponding to each variable on the X-axis [61]. The bars indicate the factors with the highest VIP scores and, thus, are the most contributory variables in class discrimination in the PLS-DA model, assuming a *p*-value > 1 as statistically significant. The statistical analysis was carried out using MedCalc for Windows, version 15.4 64 bit (MedCalc Software, Ostend, Belgium), and SIMCA-P version 13.0, (Umetrics AB, Umea, Sweden).

## 5. Conclusions

Our study highlights how the plasma profile of polyamines could be useful in discriminating CRC patients. Inflammatory indexes, such as RDW and PLT, allow the identification of fingerprints of biochemical activity in CRC. Plasma assessment of ornithine and acetylspermine, and NLR, could provide useful information regarding disease progression. High-resolution mass spectrometry combined with high-performance liquid chromatography seems promising and could become a tool to support diagnostic tumor stratification. Our data improve the understanding of the complex interconnections between polyamine metabolism and neoplastic progression, which are also influenced by the microenvironment surrounding the tumor.

## Figures and Tables

**Figure 1 ijms-24-00101-f001:**
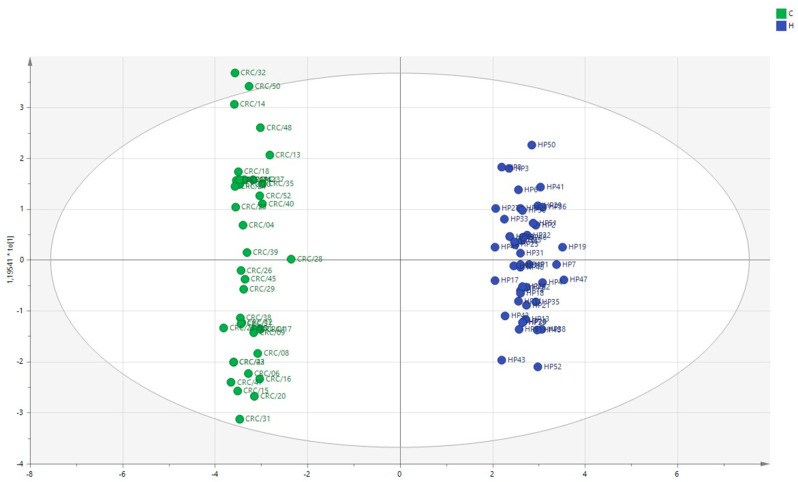
Multivariate analysis using PLS-DA method. Colorectal carcinoma group (green); healthy subject group (blue).

**Figure 2 ijms-24-00101-f002:**
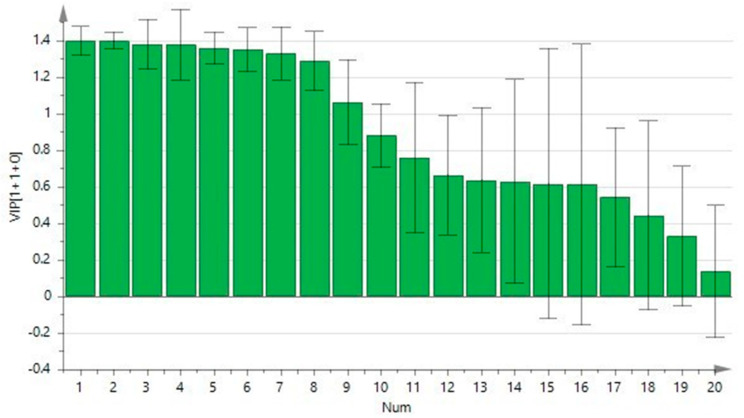
Contribution plot from the model including total VIP; peaks with positive contribution scores correspond to metabolites with higher levels. The Y-axis indicates the VIP scores corresponding to each variable on the X-axis.

**Figure 3 ijms-24-00101-f003:**
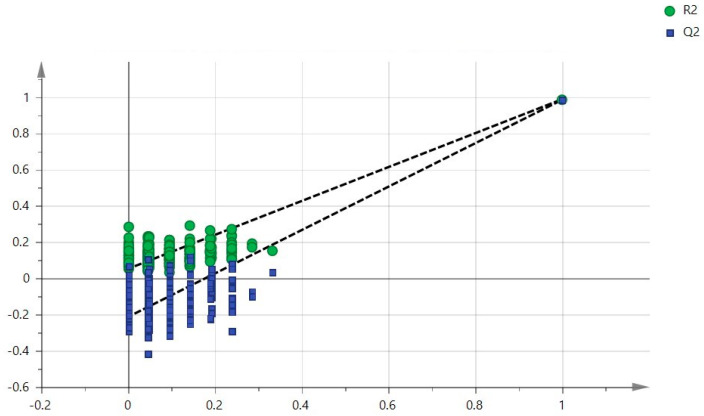
Validation method: 300-fold cross-permutation validation plot. The Y-axis represents R2 (triangles) and Q2 (circles) for the model, and the X-axis designates the correlation coefficient between the original and the permuted response data.

**Table 1 ijms-24-00101-t001:** Clinical and biochemical parameters between patients with colorectal carcinoma and healthy people.

	Cancer Patients	Healthy Patients	*p* Value
**Age**	67.74 ± 10.44	55.42 ± 5.14	*p* < 0.001
**WBC**	6.50 (5.26–7.86)	6 (5.45–7.05)	*p* = 0.325
**HGB**	12.08 ± 2.15	13.70 ± 0.89	*p* < 0.001
**RDW**	15.10 (13.42–17.00)	87.00 (84.00–89.70)	*p* < 0.001
**MCV**	85.30 (72.95–92.35)	14.60 (12.70–15.30)	*p* < 0.001
**NEUT**	3.90 (3.10–5.22)	2.13 (1.72–2.50)	*p* < 0.001
**LYMPH**	1.51 ± 0.52	3.50 ± 1.15	*p* < 0.001
**MONO**	0.41 (0.30–0.63)	0.40 (0.30–0.43)	*p* = 0.022
**PLT**	221.00 (180.00–309.75)	224.50 (203.50–264.00)	*p* = 0.905
**MPV**	7.90 (7.20–8.35)	8.45 (7.91–9.10)	*p* < 0.001
**AISI**	308.00 (142.76–518.90)	139.45 (98.81–206.05)	*p* < 0.001
**SII**	663.00 (453.23–899.14)	363.13 (277.86–500.06)	*p* < 0.001
**SIRI**	1.22 (0.67–2.07)	0.60 (0.45–0.77)	*p* < 0.001
**MLR**	0.29 (0.21–0.47)	0.17 (0.14–0.22)	*p* < 0.001
**MPR**	0.03 (0.02–0.04)	0.04 (0.03–0.04)	*p* = 0.287
**NLPR**	0.01 (0.01–0.02)	0.01 (0.01–0.01)	*p* < 0.001
**NLR**	2.50 (2.11–3.53)	1.51 (1.29–1.91)	*p* < 0.001
**dNLR**	1.70 (1.41–2.30)	1.26 (1.08–1.58)	*p* < 0.001
**PLR**	162.67 (122.46–209.84)	110.53 (91.13–138.25)	*p* < 0.001
**HGB/RDW**	0.82 ± 0.25	0.97 ± 0.12	*p* < 0.001

Age, HGB, lymphocytes, and HGB/RDW are reported as mean ± standard deviation and analyzed using the Student–Newman–Keuls test. All other values are reported as median and interquartile range (IQR) and analyzed using the Kruskal–Wallis test. A *p*-value < 0.05 is considered to be statistically significant. WBC: white blood cell; HGB: hemoglobin; RDW: red cell distribution width; MCV: mean cell volume; NEUT: neutrophils; LYMPH: lymphocytes; MONO: monocytes; PLT: platelet; MPV: mean platelet volume; AISI: aggregate index of systemic inflammation; SII: systemic immune-inflammation index; SIRI: systemic inflammation response index; MLR: monocyte to lymphocyte ratio; MPR: mean platelet volume to platelet ratio; NLPR: neutrophil to lymphocyte × platelet ratio; NLR: neutrophil to lymphocyte ratio; dNLR: derived neutrophil to lymphocyte ratio; PLR: platelet to lymphocyte ratio.

**Table 2 ijms-24-00101-t002:** Correlation between inflammatory indexes and clinical-pathological features in patients with colorectal carcinoma.

VARIABLES		INFLAMMATION INDEXEX																			
		AISI		SII		SIRI		MLR		MPR		NLPR		NLR		dNLR		PLR		HGB/RDW	
		Median (IQR)	*p* Value	Median (IQR)	*p* Value	Median (IQR)	*p* Value	Median (IQR)	*p* Value	Median (IQR)	*p* Value	Median (IQR)	*p* Value	Median (IQR)	*p* Value	Median (IQR)	*p* Value	Median (IQR)	*p* Value	Media ± SD	*p* Value
**Gender**																					
Female. *n (*%)	17 (34.0)	275.3 (122.7–506.4)	0.749	641.0 (502.0–1266.1)	0.447	0.9 (0.5–2.1)	0.540	0.2 (0.2–0.4)	0.141	0.03 (0.02–0.03)	0.152	0.01 (0.01–0.01)	0.103	2.4 (2.0–3.3)	0.713	1.8 (1.3–2.4)	0.500	165.1 (131.8–238.9)	0.254	0.8 ± 0.2	0.878
Male. *n* (%)	33 (66.0)	323.1 (154.5–537.3)		663.0 (401.8–884.2)		1.4 (0.7–2.0)		0.3 (0.2–0.5)		0.04 (0.02–0.05)		0.01 (0.01–0.02)		2.6 (2.1–3.5)		1.7 (1.4–2.2)		154.7 (107.8–200.6)		0.8 ± 0.2	
**Age at diagnosis**																					
< 65 years. *n (*%)	19 (38.0)	319.7 (156.00–584.9)	0.417	673.5 (502.0–1266.1)	0.391	1.2 (0.7–2.7)	0.659	0.2 (0.2–0.6)	0.643	0.03 (0.02–0.04)	0.730	0.01 (0.01–0.02)	0.968	2.7 (1.8–5.6)	0.826	1.7 (1.3–2.8)	0.586	166.9 (122.7–238.9)	0.502	0.8 ± 0.3	0.983
≥ 65 years. *n (*%)	31 (62.0)	293.8 (108.43–475.7)		644.3 (401.8–884.2)		1.2 (0.7–1.7)		0.3 (0.2–0.4)		0.03 (0.02–0.04)		0.01 (0.01–0.02)		2.4 (2.2–3.5)		1.7 (1.4–2.1)		154.7 (112.1–204.8)		0.8 ± 0.2	
**Site**																					
Left. *n* (%)	35 (70.0)	241.0 (114.5–446.6)	0.058	595.0 (393.8–795.1)	0.095	1.1 (0.6–2.1)	0.461	0.3 (0.2–0.5)	0.760	0.04 (0.03–0.04)	**0.002**	0.01 (0.01–0.02)	0.087	2.6 (2.0–3.8)	0.867	1.7 (1.4–2.3)	0.956	145.2 (104.9–185.6)	**0.020**	0.9 ± 0.2	**<0.001**
Right. *n* (%)	12 (24.0)	446.5 (277.0–589.4)		857.7 (520.0–1227.7)		1.2 (1.1–1.7)		0.3 (0.3–0.4)		0.02 (0.02–0.03)		0.01 (0.01–0.01)		2.3 (2.2–3.3)		1.7 (1.4–2.2)		201.3 (181.5–235.9)		0.6 ± 0.2	
**Histologic grade**																					
G1-G2. *n* (%)	39 (76.5)	300.9 (136.1–509.3)	0.590	673.5 (419.9–875.4)	0.67	1.2 (0.7–2.1)	0.702	0.3 (0.2–0.5)	0.307	0.03 (0.03–0.04)	0.120	0.01 (0.01–0.02)	**0.016**	2.7 (2.1–3.8)	0.712	1.7 (1.3–2.3)	0.967	161.1 (122.2–203.7)	0.629	0.8 ± 0.2	0.194
G3. *n* (%)	10 (19.6)	323.1 (174.3–559.2)		592.1 (480.0–1213.5)		1.2 (0.6–1.7)		0.3 (0.2–0.4)		0.03 (0.02–0.03)		0.01 (0.01–0.01)		2.3 (2.1–3.1)		1.6 (1.5–2.1)		162.7 (127.4–244.2)		0.7 ± 0.2	
**Tumor stage**																					
0-I-II. *n (%)*	24 (47.1)	300.9 (149.4–446.5)	0.302	630.5 (466.3–819.7)	0.360	1.2 (0.7–1.7)	0.481	0.3 (0.2–0.4)	0.324	0.03 (0.03–0.04)	0.961	0.01 (0.01–0.02)	0.467	2.5 (2.2–3.5)	0.549	1.6 (1.4–2.1)	0.467	160.4 (128.8–198.5)	0.751	0.8 ± 0.2	0.398
III-IV. *n* (%)	24 (47.1)	351.0 (156.0–600.4)		707.6 (474.1–1196.0)		1.3 (0.7–2.1)		0.3 (0.2–0.7)		0.03 (0.02–0.04)		0.01 (0.01–0.02)		2.9 (2.1–5.3)		1.7 (1.4–2.4)		170.5 (122.7–238.9)		0.8 ± 0.3	

SD: standard deviation; IQR: interquartile range; AISI: aggregate index of systemic inflammation; SII: systemic immune-inflammation index; SIRI: systemic inflammation response index; MLR: monocyte to lymphocyte ratio; MPR: mean platelet volume to platelet ratio; NLPR: neutrophil to lymphocyte × platelet ratio; NLR: neutrophil to lymphocyte ratio; dNLR: derived neutrophil to lymphocyte ratio; PLR: platelet to lymphocyte ratio; HGB/RDW: hemoglobin to red cell distribution width ratio.

**Table 3 ijms-24-00101-t003:** Levels of plasma polyamines and related amino acids (arginine and lysine) in patients with colorectal carcinoma and healthy people.

Polyamine	Cancer Patients	Healthy Patients	*p*-Value
**Agmatine**	24.1 ± 2.6	77.2 ± 14.5	*p* < 0.001
**Arginine**	7.6 ± 0.6	6.4 ± 1.7	*p* < 0.001
**Cadaverine**	14.2 ± 1.7	2.5 ± 1.1	*p* < 0.001
**Lysine**	1.1 (1.0–1.1)	0.7 (0.4–1.1)	*p* < 0.001
**Ornithine**	2.6 (2.6–2.7)	0.9 (0.6 −1.4)	*p* < 0.001
**Putrescine**	17.7 (16.9–18.7)	6.4 (5.6–7.1)	*p* < 0.001
**Acetylputrescine**	289.4 (279.3–299.5)	0.3 (0.2–0.6)	*p* < 0.001
**Spermidine**	129.2 (124.9–135.3)	0.9 (0.8–1.0)	*p* < 0.001
**Acetylspermidine**	10.3 (8.9–11.8)	0.4 (0.3–0.7)	*p* < 0.001
**Spermine**	7.0 ± 1.8	5.9 ± 2.1	*p* = 0.005
**Acetylspermine**	14.7 ± 1.3	2.3 ± 1.0	*p* < 0.001

Agmatine, arginine, cadaverine, spermine, and acetylspermine are reported as mean ± standard deviation and analyzed using the Student–Newman–Keuls test. All other values are reported as median and interquartile range (IQR) and analyzed using the Kruskal–Wallis test. A *p*-value < 0.05 is considered to be statistically significant.

**Table 4 ijms-24-00101-t004:** Correlation between polyamine levels and clinical-pathological features in patients with colorectal carcinoma.

			Polyamines																				
		AGMATINE		ARGININE		CADAVERINE		LYSINE		ORNITHINE		PUTRESCINE		ACETYL-PUTRESCINE		SPERMIDINE		ACETYL-SPERMIDINE		SPERMINE		ACETYL-SPERMINE	
		Media ± SD	*p* Value	Media ± SD	*p* Value	Media ± SD	*p* Value	Median (IQR)	*p* Value	Median (IQR)	*p* Value	Median (IQR)	*p* Value	Median (IQR)	*p* Value	Median (IQR)	*p* Value	Median (IQR)	*p* Value	Media ± SD	*p* Value	Media ± SD	*p* Value
**Gender**																							
Female. *n (*%)	17 (34.0)	24.0 ± 2.4	0.819	7.6 ± 0.8	0.942	13.9 ± 1.7	0.419	1.1 (1.0–1.1)	0.728	2.6 (2.6–2.6)	0.943	17.8 (17.1–18.8)	0.384	297.3 (287.7–306.2)	**0.029**	126.8 (121.2–134.6)	0.320	9.38 (8.6–11.4)	0.260	6.7 ± 1.9	0.388	14.8 ± 1.6	0.562
Male. *n* (%)	33 (66.0)	24.2 ± 2.7		7.6 ± 0.5		14.3 ± 1.7		1.1 (1.0–1.1)		2.6 (2.6–2.7)		17.6 (17.1–18.8)		285.6 (278.6–295.6)		129.2 (126.4–134.3)		10.6 (9.4–15.2)		7.2 ± 1.7		14.6 ± 1.1	
**Age at diagnosis**																							
< 65 years. *n (*%)	19 (38.0)	23.5 ± 2.2	0.194	7.7 ± 0.7	0.645	14.1 ± 1.7	0.687	1.1 (1.0–1.1)	0.764	2.6 (2.6–2.6)	0.45	17.4 (16.8–18.1)	0.105	286.7 (280.1–304.0)	0.49	130.8 (123.1–135.5)	0.834	10.1 (8.6–11.8)	0.749	7.2 ± 1.8	0.573	14.8 ± 1.7	0.619
≥ 65 years. *n (*%)	31 (62.0)	24.5 ± 2.8		7.6 ± 0.6		14.3 ±1.7		1.1 (1.0–1.1)		2.6 (2.6–2.7)		17.9 (17.4–18.9)		290.2 (278.4–299.0)		129.1 (125.3–133.7)		10.4 (9.2–11.8)		6.9 ± 1.8		14.6 ± 1.0	
**Localization**																							
Left. *n* (%)	35 (70.0)	23.7 ± 2.5	0.044	7.7 ± 0.6	0.469	14.2 ± 1.8	0.808	1.1 (1.0–1.1)	0.428	2.6 (2.6–2.6)	0.893	17.6 (17.0–18.4)	0.696	288.5 (279.4–301.0)	0.903	129.2 (125.0–134.6)	0.942	10.2 (9.0–11.8)	0.329	7.1 ± 1.7	0.856	14.7 ± 1.4	0.945
Right. *n* (%)	12 (24.0)	25.5 ± 2.7		7.5 ± 0.6		14.1 ± 1.1		1.1 (1.0–1.2)		2.6 (2.4–2.7)		18.0 (16.9–19.1)		288.5 (279.5–301.3)		129.6 (124.0–134.4)		11.0 (9.9–12.0)		7.2 ± 2.1		14.7 ± 1.0	
**Histologic grade**																							
G1-G2. *n* (%)	39 (76.5)	24.1 ± 2.8	0.776	7.6 ± 0.6	0.727	14.3 ± 1.5	0.754	1.1 (1.0–1.1)	0.188	2.6 (2.6–2.7)	**0.047**	17.8 (17.0–18.4)	0.620	286.7 (278.9–299.5)	0.234	129.8 (126.3–134.3)	0.096	10.3 (9.0–11.8)	0.551	7.1 ± 1.7	0.727	14.6 ± 1.3	0.448
G3. *n* (%)	10 (19.6)	24.3 ± 2.1		7.5 ± 0.8		14.1 ± 2.3		1.1 (1.1–1.2)		2.6 (2.4–2.6)		18.1 (17.1–19.6)		295.4 (285.2–310.8)		124.1 (122.7–130.0)		10.9 (8.9–12.0)		6.9 ± 2.0		14.9 ± 1.2	
**Tumor stage**																							
0-I-II. *n (%)*	24 (47.1)	24.7 ± 2.5	0.19	7.7 ± 0.5	0.238	14.3 ± 1.5	0.846	1.1 (1.0–1.1)	0.828	2.6 (2.6–2.7)	0.261	17.9 (17.4–18.8)	0.332	291.2 (279.5–299.5)	0.837	129.2 (125.0–133.5)	0.901	9.8 (9.0–11.8)	0.869	7.0 ± 1.7	0.893	15.1 ± 1.3	**0.009**
III-IV. *n* (%)	24 (47.1)	23.6 ± 2.8		7.5 ± 0.7		14.2 ± 1.9		1.1 (1.0–1.1)		2.6 (2.6–2.6)		17.7 (16.8–18.7)		291.8 (279.3–304.0)		129.4 (123.6–134.5)		10.3 (8.7–11.6)		7.0 ± 1.8		14.2 ± 1.1	

SD = standard deviation; IQR = interquartile range.

## Data Availability

Not applicable.

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
