# Peer review of "Quantitative Metabolomics to Explore the Role of Plasma Polyamines in Colorectal Cancer"

_ijms, 2022, doi:10.3390/ijms24010101_

Round 1

Reviewer 1 Report

The manuscript by Coradduzza and co-workers reports on how the plasma profile of polyamines could be useful in discriminating CRC patients. Authors described that Plasma assessment of ornithine and acetylspermine, as well as lymphocytes/platelets ratio could provide useful information on progression of CRC. They also demonstrated that high-resolution mass spectrometry combined with high-performance liquid chromatography could be a promising tool to support diagnostic tumor stratification. These results seem informative and interesting, however, I have several concerns, which may further improve this manuscript.    

1.     How can authors discriminate inflammatory disease, such as IBD, from CRC patients and healthy control by using inflammatory indexes? Furthermore, the inflammatory indexes may apply to all kind of tumors, as well as the plasma polyamines.

2.     This manuscript should be revised by statistical experts.

3.     The reference within recent 5 years should be cited, especial for some important reference (such as reference 1-6, 14-22, 24, 25, 31-38, 41, 42). Authors should consider to remove relevant outdated content if updated reference can unable be searched.

4.     Some of the references are either inappropriate or less authoritative.

5.     Reference 11 is unavailable.

6.   The manuscript should be edited by an English native speaker, there are many language errors and inappropriate words in the article.    

Author Response

Reviewer 1

  1. How can authors discriminate inflammatory disease, such as IBD, from CRC patients and healthy control by using inflammatory indexes? Furthermore, the inflammatory indexes may apply to all kind of tumors, as well as the plasma polyamines.

We appreciate the reviewer's comments, but all CRC patients and healthy subjects included in the study screened for this parameter and were inflammatory bowel disease (IBD)-free as defined by medical records. We specify this information in the Materials and Methods section (see lanes 367-369).. In addition, we have studied these inflammatory indices and plasma polyamines in tumors or other diseases (e.g. Prostate Cancer, Lymphoma Patients, etc.) as well as other researchers.

Wu Y, Tu C, Shao C. Inflammatory indexes in preoperative blood routine to predict early recurrence of hepatocellular carcinoma after curative hepatectomy. BMC Surg. 2021 Apr 1;21(1):178. doi: 10.1186/s12893-021-01180-9. PMID: 33794850; PMCID: PMC8017621.

Hayama T, Ozawa T, Asako K, Kondo R, Ono K, Okada Y, Tsukamoto M, Fukushima Y, Shimada R, Nozawa K, Matsuda K, Fujii S, Fukagawa T, Hashiguchi Y. Impact of Colon Cancer Location on the Prognostic Significance of Nutritional Indexes and Inflammatory Markers. In Vivo. 2021 Mar-Apr;35(2):1261-1269. doi: 10.21873/invivo.12377. PMID: 33622929; PMCID: PMC8045081.

Zou Y, Chen Z, Lou Q, Han H, Zhang Y, Chen Z, Ma Z, Shi N, Jin H. A Novel Blood Index-Based Model to Predict Hepatitis B Virus-Associated Hepatocellular Carcinoma Recurrence After Curative Hepatectomy: Guidance on Adjuvant Transcatheter Arterial Chemoembolization Choice. Front Oncol. 2021 Dec 24;11:755235. doi: 10.3389/fonc.2021.755235. PMID: 35004275; PMCID: PMC8739488.

Coradduzza D, Solinas T, Azara E, Culeddu N, Cruciani S, Zinellu A, Medici S, Maioli M, Madonia M, Carru C. Plasma Polyamine Biomarker Panels: Agmatine in Support of Prostate Cancer Diagnosis. Biomolecules. 2022 Mar 29;12(4):514. doi: 10.3390/biom12040514. PMID: 35454104; PMCID: PMC9024899.

Zinellu A, Collu C, Zinellu E, Ahmad K, Nasser M, Traclet J, Sotgiu E, Mellino S, Mangoni AA, Carru C, Pirina P, Cottin V, Fois AG. IC4: a new combined predictive index of mortality in idiopathic pulmonary fibrosis. Panminerva Med. 2022 Jun;64(2):228-234. doi: 10.23736/S0031-0808.21.04144-6. Epub 2021 Jan 26. PMID: 33496152.

  1. This manuscript should be revised by statistical experts.

We thank the reviewer for the suggestion. We would like to point out that the list of authors includes Prof. Zinellu who is an expert in statistics. The statistical analysis we use, particularly OPLS-DA, has also been validated and developed by various authors (see Leruez S, 2018; Graham SF, 2015; Bylesjö, M, 2006) and used in particular for polyamine analysis. Similar approaches to the one used in our manuscript have been in the literature for some time and previously used by us as well.  Below is a short list of articles using this methodology:

Leruez S, et al., A Plasma Metabolomic Signature of the Exfoliation Syndrome Involves Amino Acids, Acylcarnitines, and Polyamines. Invest Ophthalmol Vis Sci. 2018 Feb 1;59(2):1025-1032. doi: 10.1167/iovs.17-23055.

Graham SF, et al., Untargeted metabolomic analysis of human plasma indicates differentially affected polyamine and L-arginine metabolism in mild cognitive impairment subjects converting to Alzheimer's disease. PLoS One. 2015 Mar 24;10(3):e0119452. doi: 10.1371/journal.pone.0119452. eCollection 2015.

Bylesjö, M.; Rantalainen, M.; Cloarec, O.; Nicholson, J.K.; Holmes, E.; Trygg, J. OPLS discriminant analysis: combining the strengths of PLS-DA and SIMCA classification. J Chemom 2006, 20, 341–351, doi:10.1002/CEM.1006.

Coradduzza D, et al., Role of Polyamines as Biomarkers in Lymphoma Patients: A Pilot Study. Diagnostics (Basel). 2022 Sep 4;12(9):2151. doi: 10.3390/diagnostics12092151. PMID: 36140552; PMCID: PMC9497571.

  1. The reference within recent 5 years should be cited, especial for some important reference (such as reference 1-6, 14-22, 24, 25, 31-38, 41, 42). Authors should consider to remove relevant outdated content if updated reference can unable be searched.

According to the Reviewer’s comment, we revised the references and modified or deleted when appropriate. Reference 1 has been updated with GLOBOCAN 2020, but the references 2,3,4,5,6 are widely used to define CRC molecularly and clinically. References 36,37,38 are specific because analyzed certain preneoplastic o early colorectal lesions and also validate our results. Finally, to our knowledge, references 41 and 42 are useful because no other studies on agmatine and colorectal cancer have been conducted.

  1. Some of the references are either inappropriate or less authoritative.

According to the Reviewer’s comment, we completely revised the references and modified or eliminated when necessary.

  1. Reference 11 is unavailable.

The reference 11 is available on PUBMED (PMID: 20420949) and easily identified by the DOI: 10.1053/j.gastro.2010.01.058, which is present in references.

  1. The manuscript should be edited by an English native speaker, there are many language errors and inappropriate words in the article.

According to the Reviewer’s comment, the manuscript was proofread, and experienced scholarly writers have edited this manuscript. The final version has also been checked by a Native Language Speaker.

Reviewer 2 Report

The article aimed  to test whether the levels of circulating polyamines in plasma and the inflammation indexes can discriminate colorectal cancer (CRC) patients from non-CRC people, and to relate the clinical-pathological features of CRC to biochemical signatures. CRC is one of the major clinical, public health and economic problem and development of non-invasive screening tests like based on plasma analysis is needed. Ultra-high resolution liquid chromatography and mass spectrometry methods were used in this study. Plasma assessment of ornithine and acetylspermine, as well as lymphocytes/platelets might be helpful in diagnostics. 

This is a retrospective study based on 50 patient-derived samples. Preliminary results are promising and should be further tested in new, bigger, and heterogeneous cohorts. In Discussion section the author's prepared proper analysis of limitation of data presented and proposed continuation of research.

Minor issue:

In most of main and supplementary tables and figures commas "," were used which must be transferred to dots "." e.g. 12,08 +/- 2,15 --> 12.08 +/- 2.15, or 0 Y axis in many supplementary plots.

Author Response

Reviewer 2

1.In most of main and supplementary tables and figures commas "," were used which must be transferred to dots "." e.g. 12,08 +/- 2,15 --> 12.08 +/- 2.15, or 0 Y axis in many supplementary plots.

We apologize for the inaccuracy. Accordingly with the reviewer’s comment, we modified the comma with point in tables and figures.

Reviewer 3 Report

Coradduzza et al in their manuscript "Quantitative metabolomics explores the role of plasma polyamines in colorectal cancer" propose an interesting association between CRC and plasma polyamine levels. The manuscript has a solid introduction with insightful discussion on the polyamine levels in CRC pathogenesis. 

This reviewer has only a few minor editorial concerns:

1. Line 97: "The media age at diagnosis" : Median age

2. Lines 109 and 111: Please write "healthy subjects" 

A few questions could be briefly addressed in the discussion:

1. Given the known association between gut microbial dyshomeostasis  in CRC pathogenesis, and the role of microbiota in gut  polyamaine levels, the authors may consider add a few sentences in line of their findings. 

2. What is known about the plasma polyamine level alterations in other cancers? 

Author Response

Reviewer 3

  1. Line 97 : "The media age at diagnosis" : Median age

Following reviewer comment, we eliminate “at diagnosis”

  1. Lines 109 and 111: Please write "healthy subjects"

We modified the phrase as suggested

  1. Given the known association between gut microbial dyshomeostasis in CRC pathogenesis, and the role of microbiota in gut polyamine levels, the authors may consider add a few sentences in line of their findings.

According to the reviewer comment, we discussed the role of gut microbiota and its polyamine synthesis in the pathogenesis of CRC (see lanes 290-299).

  1. What is known about the plasma polyamine level alterations in other cancers?

According to the reviewer comment, we discussed alterations of polyamine levels in other cancers (see lanes 283-289).

Round 2

Reviewer 1 Report

The revised manuscript has addressed most of my concerns raised based on the previous version, and has significantly improved its quality. However, authors still can not discriminate inflammatory disease, such as IBD, from CRC patients by using inflammatory indexes, inflammatory disease cohort may should also be verified by this inflammatory indexes.

Author Response

The revised manuscript has addressed most of my concerns raised based on the previous version, and has significantly improved its quality. However, authors still can not discriminate inflammatory disease, such as IBD, from CRC patients by using inflammatory indexes, inflammatory disease cohort may should also be verified by this inflammatory indexes.

We added between lines 367-370 the following sentence:

Coexistence of clinical or histopathological findings inflammatory bowel disease (IBD; i.e. ulcerative colitis or Crohn's disease) was an exclusion criterion from the study; in particular no case of CRC arose in IBD and no patient in the healthy group was affected by chronic bowel disease.

We emphasize that during screening and enrollment of our CRC patients, we checked inflammatory indices and checked for the presence of IBD in our cohort of patients and healthy subjects.

We apologize for not highlighting this aspect more clearly in our previous versions of the manuscript, but during recruitment this parameter was always taken into great consideration.

We emphasize again that our samples are free of IBD.

We hope that we have comprehensively answered the reviewer's comment and would be available for further clarification if it is needed.